# Testing both affordability-availability and psychological-coping mechanisms underlying changes in alcohol use during the COVID-19 pandemic

Orla McBride[1]*, Eimhear Bunting[1], Oisín Harkin[1], Sarah Butter[1,2], Mark Shevlin[1], Jamie Murphy[1], Liam Mason[3], Todd K. Hartman[4], Ryan McKay[5], Philip Hyland[6], Liat Levita[2], Kate M. Bennett[7], Thomas V. A. Stocks[2], Jilly Gibson-Miller[2], Anton P. Martinez[2], Frédérique Vallières[8], Richard P. Bentall[2]

**1** Ulster University, Coleraine, Northern Ireland, **2** University of Sheffield, Sheffield, England, **3** University College London, London, England, **4** University of Manchester, Manchester, England, **5** Royal Holloway, University of London, Egham, England, **6** Maynooth University, Maynooth, Republic of Ireland, **7** University of Liverpool, Liverpool, England, **8** Trinity College Dublin, Dublin, Republic of Ireland

* o.mcbride@ulster.ac.uk

**Data Availability Statement:** All data files and analytic code are available from the Open Science Framework https://osf.io/w3anh/.

## Abstract

Two theoretical perspectives have been proffered to explain changes in alcohol use during the pandemic: the 'affordability-availability' mechanism (i.e., drinking decreases due to changes in physical availability and/or reduced disposable income) and the 'psychological-coping' mechanism (i.e., drinking increases as adults attempt to cope with pandemic-related distress). We tested these alternative perspectives via longitudinal analyses of the COVID-19 Psychological Consortium (C19PRC) Study data (spanning three timepoints during March to July 2020). Respondents provided data on psychological measures (e.g., anxiety, depression, posttraumatic stress, paranoia, extraversion, neuroticism, death anxiety, COVID-19 anxiety, intolerance of uncertainty, resilience), changes in socio-economic circumstances (e.g., income loss, reduced working hours), drinking motives, solitary drinking, and 'at-risk' drinking (assessed using a modified version of the AUDIT-C). Structural equation modelling was used to determine (i) whether 'at-risk' drinking during the pandemic differed from that recalled before the pandemic, (ii) dimensions of drinking motives and the psychosocial correlates of these dimensions, (iii) if increased alcohol consumption was predicted by drinking motives, solitary drinking, and socio-economic changes. The proportion of adults who recalled engaging in 'at-risk' drinking decreased significantly from 35.9% pre-pandemic to 32.0% during the pandemic. Drinking to cope was uniquely predicted by experiences of anxiety and/or depression and low resilience levels. Income loss or reduced working hours were not associated with coping, social enhancement, or conformity drinking motives, nor changes in drinking during lockdown. In the earliest stage of the pandemic, psychological-coping mechanisms may have been a stronger driver to changes in adults' alcohol use than 'affordability-availability' alone.

**Funding:** The COVID-19 Psychological Research Consortium (C19PRC) Study (authors OMB, MS, JM, LM, TKH, LL, KMB, JGM, RPB) is funded by the UK Research and Innovation Economic and Social Research Council (UKRI ESRC) grant number ES/V004379/1. https://gtr.ukri.org/projects?ref=ES%2FV004379%2F1 The sponsor or funder played no role in the study design, data collection and analysis, decision to publish, or preparation of the manuscript.

**Competing interests:** The authors have declared that no competing interests exist.

## Introduction

A defining feature of the COVID-19 pandemic has been the widespread use of mandatory 'stay-at-home' directives to limit the transmission of the virus [1]. Whilst the public health benefit afforded by such directives in reducing rates of infections and mortality became apparent early in the pandemic [2], they also dramatically restricted the spaces that people occupied and, importantly, the contexts in which they consumed alcohol [3]. Clay and Parker [4] rightly cautioned that "*the potential public health effects of long-term isolation* [due to the COVID-19 pandemic] *on alcohol use and misuse are unknown*" (p.e259). Attempts have been made to assimilate evidence from studies conducted during other periods of large-scale, population-level social, economic, and public health uncertainty, such as the 2002–03 SARS pandemic, the Great Economic Recession of 2007–09, and other traumatic events or natural disasters (e.g., Twin Towers, Hurricane Katrina) [5, 6], to theorise as to the most likely impact of the COVID-19 pandemic on adults' alcohol use. Two broad scenarios have been proposed.

The first is that, for some groups of the population, alcohol consumption will decrease, largely due to a lack of physical availability (i.e., closure of on-licence premises and restrictions on off-licence sales) and/or issues relating to affordability (i.e., less disposable income due to unemployment and/or reduced working hours)–the so-called 'affordability-availability' mechanism [7]. The second is that, for others, alcohol use will increase in both quantity and frequency as adults struggle to cope with stress and negative affect experienced as a result of the pandemic—the so-called 'psychological-coping' mechanism [7]. Given the sudden and widespread impact of the pandemic on socio-economic activities, Rehm and colleagues [8] suggested evidence in support of the 'affordability-availability' mechanism should be stronger in the earliest phases of the pandemic. However, they also suggested that a relaxation of alcohol control measures and increasing personal distress relating to the protracted pandemic could lead to an increase in levels of alcohol use and associated harmful consequences in the longer term, which is consistent with concerns raised elsewhere [9].

Evidence from studies testing these opposing mechanisms has begun to emerge. With respect to the 'affordability-availability' mechanism, Kilian, Rehm [7] analysed cross-sectional data of changes in adults' self-reported drinking frequency, quantity, and heavy episodic drinking events, across 21 countries between April and August 2020. Alcohol use decreased, on average, in all countries except for in Ireland and the UK (where drinking, on average, remained unchanged or increased, respectively). In other research, a descriptive analysis of alcohol use patterns identified using data from the Alcovision surveys revealed that there were no significant changes in alcohol consumption levels or frequency of drinking in Scotland during the second quarter of 2020 (compared to the same period in 2019), while overall levels of alcohol consumption in England declined [10].

These observed curtailments in England occurred despite increases in off-trade consumption, which were seemingly offset by reductions in on-trade drinking during the 2020 lockdown. Indeed, despite initial concerns [11], excessive alcohol purchasing for off-trade consumption does not appear to have been widespread in the UK during the pandemic. For example, Anderson, Llopis [12] reported that, during the first half of 2020, although excessive alcohol purchasing occurred among households characterised by higher incomes and younger adults, overall, British households did not purchase more alcohol than normal during this period when adjusting for what typically would have been purchased (and consumed) in on-licenced premises.

Support for the 'affordability-availability' mechanism may be undermined by prevalence estimates from surveys conducted in the UK. For example, Garnett, Jackson [13] reported that, during 21 March and 4 April 2020, 48.1% of UK adult drinkers reported drinking

similarly in the past week to their usual pre-pandemic levels, whereas 25.7% and 26.2% of adult drinkers reported drinking less or more than usual, respectively. Similar estimates have been reported elsewhere in the UK [14] and beyond [15].

Attempts to characterise drinkers who have increased or decreased their drinking as a result of the pandemic have also produced mixed findings. Some studies have reported changes in alcohol consumption by gender (males; [13], females; [16]), age [13, 16], income [13], concerns about COVID-19 restrictions or contracting the virus, key-worker status [13], pre-pandemic heavy drinking levels [17] and stress levels [16].

Alternatively, and in support of the 'psychological-coping' mechanism hypothesis, drinkers consuming more than usual were typically identified as being higher income earners, having higher levels of anxiety and/or depressive symptoms, feeling stress about finances or becoming infected/seriously ill with COVID-19, working from home, and caring for children at home [13–15, 18–21].

Null findings between psychological distress and alcohol consumption have also been reported. Villadsen, Patalay [22] analysed data from four UK cohort studies to compare pre-pandemic patterns of psychological distress and health behaviours to those occurring during the pandemic between March and September 2020. High risk drinking levels, defined as exceeding 12 drinks per week or 5+ drinks per session, declined from 19.1% pre-pandemic to 16.9% during the first lockdown in May 2020 and increased to 20.7% in September 2020. Associations between psychological distress and alcohol intake were weak; prevalence of anxiety or depression among adults who engaged in high-risk drinking pre-pandemic increased by 2.4% by May 2020; however, a decline of 2.9% was recorded by September 2020.

The aforementioned studies provide moderate, albeit mixed, support for the proposed 'affordability-availability' and 'psychological-coping' mechanisms for changes in alcohol use during the pandemic. A handful of studies have begun to investigate how important individual-level psychological factors, for example drinking motives [23], that might influence the overlapping pathways between psychological distress and changes in socio-economic factors, and alcohol use, during this period.

For example, in a sample of relatively high-income Canadian adult drinkers, Wardell, Kempe [24] investigated the role of coping motives in understanding pandemic-related increases in alcohol use. The study revealed that higher levels of depressive symptoms (but not levels of health-related anxiety), lower social connectedness, and having dependants under 18 years of age living at home were associated with higher levels of drinking to cope motives and, as a result, higher levels of alcohol consumption and alcohol-related problems early in the pandemic (i.e., between April and May 2020). No significant associations between gender, race/ethnicity, age, annual income, home working or home-related stressors, and coping motives or changes in alcohol consumption were observed, and notably, although income loss during the early stages of the pandemic was directly associated with increased alcohol consumption, the association was not mediated by coping motives. This finding runs counter to the hypothesis that individuals who lose income during the pandemic may increase alcohol consumption as a means of coping with the stress of having a reduced income. The authors speculated that this finding might be explained, in part, by the relatively high income status of respondents who, when faced with reduced income at the earliest stages of the pandemic, did not have to resort to 'drastic' income-saving behaviour changes (e.g., reducing their purchasing of alcohol for at-home consumption). Finally, the study revealed that whilst living alone was not associated with drinking to cope motives, and drinking to cope motives were not associated with solitary drinking, solitary drinking was associated with unique variance in alcohol problems over and above increased alcohol use and coping motives [24]. Thus, although these findings suggest that solitary drinking during the pandemic may be largely due to situational factors (i.e.,

stay-at-home orders) as opposed to drinking to cope with negative affect, solitary drinking remains a risk factor for increased alcohol-related harm during the pandemic [25], and hence, further research in this area is required [26, 27].

In this study, we seek to contribute to this emerging evidence base through analyses of data from a representative longitudinal panel study (the COVID-19 Psychological Research Consortium (C19PRC) Study), collected between March and July 2020 in the UK. The study had four objectives.

First, we estimated the extent to which 'at-risk' drinking, and overall drinking levels, among adult drinkers changed during the first four months of lockdown when compared to their (recall of) pre-pandemic drinking levels. Given the broad range of prevalence estimates of alcohol use reported in similar surveys, and the inconsistencies across studies in the measures used to assess drinking patterns, we made no specific hypothesis in relation to this objective.

Second, we tested hypotheses about the relationship between a range of motives for drinking during the lockdown–namely coping, social enhancement, and conformity–and (1) an array of *mental health conditions and experiences* (i.e., major depressive disorder, generalized anxiety disorder, post-traumatic stress disorder (PTSD), and paranoia), and *psychological factors* including personality (extraversion and neuroticism), COVID-19 related anxiety, loneliness, resilience, intolerance of uncertainty, and death-related anxiety; and (2) a range of *socioeconomic factors*, in particular, over-purchasing of alcohol and changes in work hours and loss of income, during the earliest stages of the pandemic. Considering drinking motives other than coping is an important, yet neglected, area of investigation [26]. It is possible that using alcohol during the pandemic may have supported some individuals to endure social isolation. Specifically, in the earliest stages of the pandemic, an ethos of being '(virtually) in this together' was suggested [28], and governments actively encouraged citizens to stay socially connected virtually, whilst remaining physically apart. Collective consumption of alcohol during unusual or unfamiliar social contexts may help create a sense of intimacy and connection [29]. Consistent with previous research [30], we hypothesised that experiences of mental health difficulties (e.g., anxiety and/or depression) and poorer psychological wellbeing (e.g., lower levels of resilience, higher levels of neuroticism, intolerance of uncertainty, and death anxiety), would be most strongly associated with drinking to cope motives, followed by drinking to conform motives, but not necessarily with social enhancement drinking motives. We also hypothesised that reduced working hours and loss of income would be associated with coping motives, but not with drinking-to-conform or social enhancement motives. Given the growing, yet mixed, body of evidence in this area, we expected to obtain significant associations between key sociodemographic characteristics (e.g., being female, younger, living with dependants under 18 years, a key-worker, high income earner, etc.) and coping motives, but we made no specific hypotheses relating to drinking to conform or social enhancement motives.

Third, further testing the 'psychological' mechanism, we hypothesised that drinking motives would directly predict greater: (i) solitary drinking and (ii) drinking during the lockdown, whilst controlling for the effect of selected respondent characteristics on drinking motives. We specifically focused on solitary drinking to the exclusion of other drinking contexts (e.g., at home or online drinking with others) because the weight of existing evidence indicated that solitary drinking is a strong predictor of increased alcohol consumption and related harm [31].

Fourth to test further the 'affordability-availability' mechanism, we estimated the direct relationship among income loss, reduced working hours, and drinking during the lockdown, independent of pathways predicting increased drinking for those who reported income losses and reduced working hours via higher levels of drinking to cope motives.

## Materials and methods

### Study

The COVID-19 Psychological Research Consortium (C19PRC) Study is a longitudinal, nationally representative, internet-based panel survey of adults living in the UK which was designed to monitor and assess the psychological and socio-economic impacts of the pandemic on the lives of ordinary citizens [32]. At the time this study was conducted, three waves of data had been collected: Wave 1 (23–28 March 2020; N = 2025); Wave 2 (22 April-1 May 2020; N = 1406, 69.4% retention from Wave 1); and Wave 3 (9–23 July 2020; N = 1166; 57.9% retention from Wave 1). Detailed methodological accounts of the study are available elsewhere [32, 33].

### Measures

**Alcohol use (Wave 2).** Respondents were identified as current drinkers if they answered 'yes' to the question '*Do you currently drink alcohol nowadays, including drinks you brew or make at home*? (see Table 1 for socio-demographic characteristics of current drinkers).

**Table 1. Sociodemographic characteristics of current drinkers in the COVID-19 Psychological Research Consortium (C19PRC) Study (N = 1406; Wave 2, April-May 2020).**

| Wave 1 (baseline) respondent characteristics (March 2020) | | Current drinkers (N = 944) N (Weighted %) |
|---|---|---|
| Gender[1] | Male | 532 (56.5%) |
| | Female | 410 (43.5%) |
| Age group (years) | 18–24 years | 46 (4.9%) |
| | 25–34 years | 118 (12.5%) |
| | 35–44 years | 158 (16.7%) |
| | 45–54 years | 212 (22.5%) |
| | 55–64 years | 219 (23.2%) |
| | 65+ years | 191 (20.2%) |
| 2019 household income | ≤£15.490 | 142 (15.0%) |
| | £15,491-£25,340 | 153 (16.2%) |
| | £25,341-£38,740 | 185 (19.6%) |
| | £38,741-£57,903 | 232 (24.6%) |
| | ≥£57,931 | 232 (24.6%) |
| Economic activity | Employed (full or part-time) | 598 (63.3%) |
| | Other | 346 (36.7%) |
| Ethnicity | White | 894 (94.7%) |
| | Other | 50 (5.3%) |
| Birthplace/Growing up | Born/Grew up in UK | 892 (94.5%) |
| | Elsewhere | 52 (5.5%) |
| Place of residence | Suburb/Town/Rural | 194 (20.6%) |
| | City | 750 (79.4%) |
| Educational attainment | Post-secondary education | 607 (64.3%) |
| | Did not attend post-secondary education | 337 (35.7%) |
| Household characteristics | Lone adult household | 195 (20.7%) |
| | Other | 749 (79.3%) |
| | Children under 18 years living in household | 128 (27.7%) |
| | Other | 334 (72.3%) |

Note.

[1] The sampling weight variable did not account for 'other gender' category and therefore weighted frequencies for respondents in this category (n = 2) are not presented.

**Alcohol dependence and at-risk drinking (Waves 2 and 3).** Current drinkers completed an adapted version of the 3-item AUDIT-C. The AUDIT-C has good specificity and sensitivity for detecting alcohol dependence and at-risk drinking in the general population [34]. At Wave 2, the AUDIT-C questions were phrased as follows: '*Prior to the lockdown (23$^{rd}$ March 2020)*: (1) *how often did you have a drink containing alcohol*? (response categories ranged from 1 'never' to 5 '4 or more times a week'); (2) *how many drinks containing alcohol did you have on a typical day when you were drinking*? (response categories 1 '1–2 drinks' to 5 '10 or more'; and (3) *how often did you have six or more alcohol drinks on one occasion*? (response categories 1 'never' to 5 'almost daily'. At Wave 3, the lead statement read '*During lockdown (in the last four months)*', with the same follow-up statements and response options as Wave 2. AUDIT-C total scores and the cut-off threshold for 'at-risk' drinking (score > = 5) were used in the analysis (see Analytic plan).

**Drinking motives (Wave 2).** Current drinkers answered 15 statements which assessed their motives for drinking alcohol in the past week (i.e., four weeks into the first national lockdown in the UK). Statements were adapted from three conceptual motive dimensions (Cooper et al. [35], as follows: *Coping* (e.g., how often do you drink to forget your worries; *Social Enhancement* (e.g., how often do you drink to have fun?); and *Conformity* (e.g., how often do you drink because you felt pressured to by your family and friends). Responses were collected on a 3-point scale (1 'never/almost never', 2 'sometime', and 3 'almost always/always'), and recoded as binary indicators representing yes/no in the previous week (see Table 3 for statement details).

**Solitary drinking (Wave 2).** Respondents were asked the extent to which they drank alcohol in the past week in the following settings: (1) on your own, in your house/garden; (2) with someone else in your own house/garden; (3) with family/friends online (e.g., WhatsApp, group Zoom/Skype); and (4) in public (e.g., outside your house/garden). Responses were rated on a 4-point frequency scale ranging from 1 'never' to 4 'always'. A single binary indicator was derived to represent 'Only drank on your own, in your house/garden' vs. 'all other drinking contexts'.

**Alcohol purchasing (Wave 2).** Respondents reported the extent to which they increased their purchasing of alcohol in the weeks before the survey on a five-point Likert scale ranging from (1) 'not at all' to (5) 'very considerably'.

**Changes in employment and household income during the pandemic (Wave 2).** Respondents were asked: *Since the lockdown, have you: (1) continued to work normal hours; (2) worked more hours; (3) worked reduced hours; (4) been placed on the Government 'furlough scheme'; or (5) stopped working for the time being*. Categories 3–5 were combined to create a binary indicator representing 'reduced working hours by May 2020' vs. other. Respondents estimated the percentage change in their monthly household income, compared to the average monthly income before the pandemic, on a visual slider scale centred at 0 and ranged from 100% (decrease) on the left-hand side to 100% (increase) on the right-hand side. A binary indicator was created to reflect 'reduced income by May 2020, -1% to -100%) vs. other.

**Depression-anxiety (Wave 1).** The Patient Health Questionnaire Anxiety-Depression Scale (PHQ-ADS) [36] is a 16-item composite measure of depression and anxiety. Respondents were asked how often, over the past 2 weeks, they were bothered by depressive (nine items) and anxiety (seven items) symptoms. Responses are scored on a four-point Likert scale (0 'not at all' to 3 'nearly every day'). Scores range from 0–48, with higher scores indicating higher levels of anxiety-depression symptomology. Moderate severity (20–48) was used to identify caseness, and scores from the PHQ-ADS have been found to demonstrate high internal reliability, as well as good convergent and construct validity in clinical samples [36, 37].

**COVID-19 posttraumatic stress disorder (PTSD) (Wave 1).** The International Trauma Questionnaire (ITQ) [38] is a self-report measure of ICD-11 PTSD. Participants completed the ITQ as follows: "*. . .in relation to your experience of the COVID-19 pandemic, please read each item carefully, then select one of the answers to indicate how much you have been bothered by that problem in the past month*". PTSD symptoms are accompanied by three items measuring functional impairment. Items are answered on a 5-point Likert scale, ranging from 0 (Not at all) to 4 (Extremely) with possible PTSD scores ranging from 0 to 24. A score of $\geq 2$ (Moderately) is considered 'endorsement' of that symptom. A PTSD diagnosis requires traumatic exposure, and at least one symptom to be endorsed from each PTSD symptom cluster (Re-experiencing, Avoidance, and Sense of Threat), and endorsement of at least one indicator of functional impairment. The psychometric properties of the ITQ scores have been demonstrated in multiple general population [38, 39] and clinical and high-risk samples [40–42].

**Paranoia (Wave 1).** Respondents completed the persecution subscale of the *Persecution and Deservedness Scale* (PaDS) [43], which is designed for use with both clinical and population samples [43, 44]. Participants rated their agreement on a 5-point scale with statements such as "I'm often suspicious of other people's intentions towards me" and "You should only trust yourself." Response options ranged from 1 = strongly disagree to 5 = strongly agree. Scale reliability for the five items was very good ($\alpha = 0.84$) in a previous epidemiological study of UK citizens [45]. Total scores were used in the analysis.

**Personality (Wave 1).** Extraversion and neuroticism were assessed using the *Big-Five Inventory (BFI-10)* [46], which contains two items per personality construct. Total scores were used in the analysis.

**Loneliness (Wave 1).** The three-item Loneliness Scale [47] is specifically designed for use in large-scale population surveys [47]. Respondents indicated how often they felt: (1) that they lacked companionship; (2) left out; and (3) isolated from others. Responses were scored on a 3-point scale (hardly ever, sometimes, or often). A threshold of $> = 6$ was used to characterise respondents as lonely [48].

**Resilience (Wave 1).** Respondents completed the 6-item *Brief Resilience Scale (BRS)* [49]. Items include 'I tend to bounce back quickly after hard times' and ' I have a hard time making it through stressful events'. Items were scored on a 5-point Likert scale ranging from 1 'strongly disagree' to 5 'strongly agree', with items 2,4 and 6 reverse coded. The BRS has demonstrated construct, convergent, and discriminant validity in the general population [50, 51]. Total scores were used in the analysis.

**Death anxiety (Wave 1).** Attitudes towards death were assessed using the 17-item *Death Anxiety Inventory (DAI)* [52]. The DAI measures four death-related anxiety factors (labelled as death acceptance, externally generated death anxiety, death finality, and thoughts about death) with items such as 'I get upset when I am in a cemetery'. Responses were scored on a 5-point Likert scale ranging from 1 'totally disagree' to 5 'totally agree' [52]. Total scores were used in the analysis.

**COVID-19 related anxiety (Wave 1).** Respondents' degree of specific anxiety about the COVID-19 pandemic was assessed using a visual slider scale, ranging from 0 'not at all anxious' on the left-hand side to 100 'extremely anxious' on the right-hand side.

**Socio-demographic characteristics (Wave 1).** Gender (males vs. females); age (18–24 years olds vs. 25–34 years, 35–44 years, 45–54 years, 55–64 years, and 65+ years groups); 2019 household income ($\leq$ £15,490 per annum vs. £15,491-£25,340, £25,341-£38,740, £38,741-£57,903, and $\geq$£57,931 bands); ethnicity (White vs. other); and household composition (living alone vs. other; children <18 years living in household vs. other).

## Ethical approval

The C19PRC Study received ethical approval from the University of Sheffield's School of Psychology Ethics Committee (Reference number 033759) [32, 33]. As outlined in our Consortium's detailed methodological reports [32, 33], survey participants (all aged 18 years or older) provided informed electronic consent (tick box) prior to commencing the survey indicating that they were informed: (i) that their data would be treated in confidence, that geolocating would be used to determine the area in which they lived (in conjunction with their residential postcode stem), and of their right to terminate participation at any time; (ii) that some topics in the survey might be sensitive or distressing (e.g., self-harm/suicide content); (iii) how their data would be stored and analysed by the research team; and (iv) that they may be contacted in the future to participate in future waves of this longitudinal survey.

## Analytic plan

A structural equation modelling (SEM) framework was used to test the study aims.

**Aim 1.** The proportion of drinkers meeting the threshold for 'at-risk' drinking, as well as mean AUDIT-C scores, relating to the retrospective recall of the pre-pandemic period (assessed at Wave 2) and during the first four months of lockdown (assessed at Wave 3) were estimated. Differences in proportions and means between the two time periods was assessed by specifying and estimating two SEMs for each outcome: (a) a null or 'constrained' model (H0) is specified where the two proportions (or means) are constrained to be equal across the Waves and the variances and covariances are freely estimated; and (b) an alternative or 'unconstrained' model (H1) is specified and estimated so that the equality constraint on the two proportions (or means) is freely estimated. The H0 and H1 models differ by one degree of freedom and improvement in model fit can be tested using a loglikelihood ratio test (LRT), which is distributed as a chi-square, using scaling correction factors for model estimates obtained using a robust maximum likelihood estimate (MLR). The MLR, which uses all available data to generate parameter estimates (and does not discard incomplete cases, nor imputes missing data) [53], was applied for all analyses unless otherwise specified.

A non-statistically significant chi-square test means that the H1 model is not a superior fit when compared to H0, and therefore the H0 (i.e., no change in alcohol use outcome between Waves) is the preferred model. Goodness of model fit was also assessed using standard information criteria, Akaike Information Criterion (AIC); Bayesian Information Criterion (BIC), and the sample-size adjusted BIC (ssa-BIC), with lower values on each criterion indicative of superior statistical fit. Fig 1 outlines the path diagrams for Aims 2–4.

**Aim 2.** There were two stages to this aim. **Stage 1:** First, confirmatory factor analysis (CFA) was used to assign the 15 observed binary drinking motive indicators to three latent variables reflecting *Coping*, *Social Enhancement*, and *Conformity* drinking motives (Fig 1). This model was estimated using the weighted-least square means and variances (WLSMV) estimator, which is suitable for analysing categorical outcome variables, and the sampling weight for this survey wave [32] was applied to deal with attrition. WLSMV offers traditional fit indices including the chi-square test, Comparative Fit Index (CFI), the Tucker-Lewis Fit Index (TLI) [54], and the root-mean square error of approximation (RMSEA) [55]. A non-significant chi-square test, as well as values of >0.95 on the CFI and TLI, and <0.06 on the RMSEA are all indicative of good model fit. Factor loadings and factor correlations were inspected to ensure that each drinking motives dimension was represented by a salient indicator ($\lambda$ >.40). Factor scores for the three latent drinking motive variables were saved for the next phase of the analysis. **Stage 2:** Second, a path model was specified and estimated to test

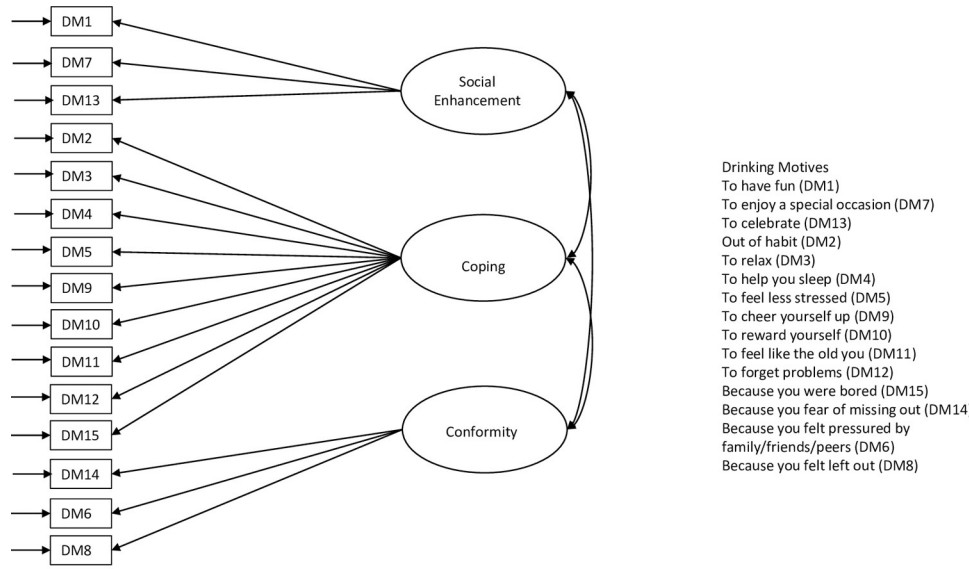

Aim 2 – Stage 1

**Fig 1. Path diagram for research aim 2 (stage 1) testing affordability-availability and psychological-coping mechanisms underlying changes in alcohol use during the COVID-19 pandemic.**

the relationships between the *Coping, Social Enhancement and Conformity* factor scores and the respondent characteristics (Fig 2).

Next, the path model was extended to relate the factor scores to two observed outcome variables to test key study hypotheses: solitary drinking (Wave 2; Aim 3) and AUDIT-C score during lockdown (Wave 3; Aim 4).

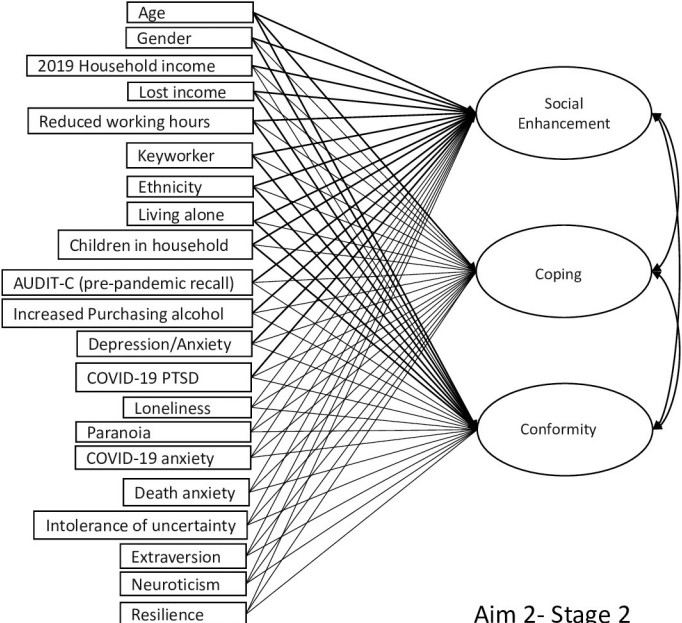

Aim 2- Stage 2

**Fig 2. Path diagram for research aim 2 (stage 2) testing affordability-availability and psychological-coping mechanisms underlying changes in alcohol use during the COVID-19 pandemic.**

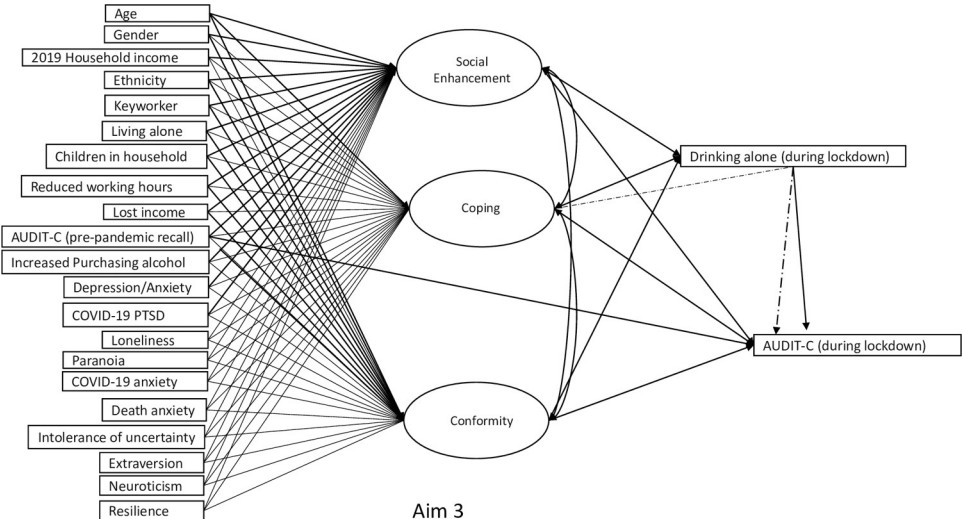

**Fig 3. Path diagram for research aim 3 testing affordability-availability and psychological-coping mechanisms underlying changes in alcohol use during the COVID-19 pandemic.**

**Aim 3.** A pathway between pre-pandemic drinking levels and drinking levels during lock-down was estimated first. Next, the AUDIT-C score (Wave 3) variable was regressed on experiences of solitary drinking and the drinking motive factor scores to test the hypotheses that solitary drinking and drinking motives directly predicted increase alcohol consumption during lockdown. Pathways between the drinking motive factors scores and solitary drinking were also estimated to test the hypothesis that only '*Coping*' would predict solitary drinking during lockdown. An indirect path between the '*Coping*' and the AUDIT-C score via solitary drinking tested the hypothesis that coping motives increased drinking during lockdown by the adult engaging more frequently in solitary drinking (Fig 3).

**Aim 4.** The model estimated at Aim 3 was extended to include additional pathways between income loss and reduced working hours and AUDIT-C scores (Fig 4).

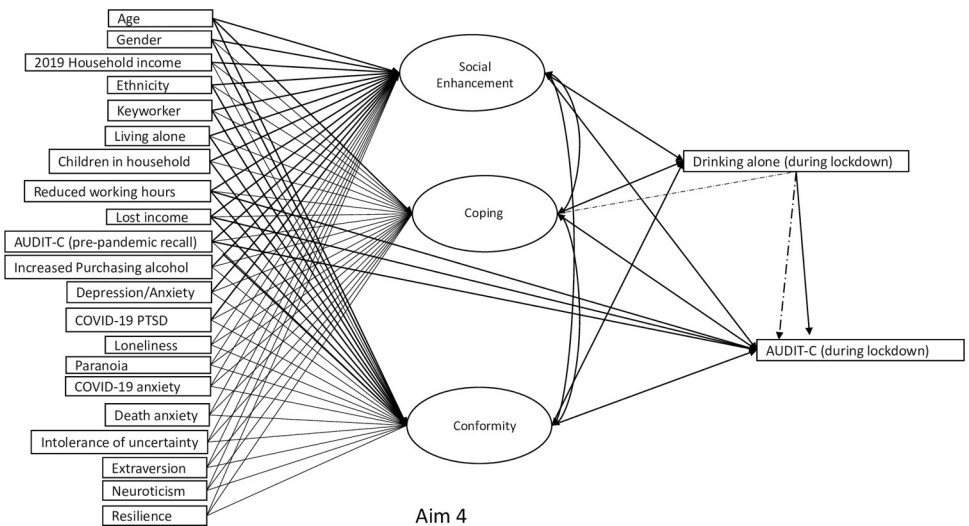

**Fig 4. Path diagram for research aim 4 testing affordability-availability and psychological-coping mechanisms underlying changes in alcohol use during the COVID-19 pandemic.**

For Aims 3–4, the AIC, BIC, and ssa-BIC are reported to assess model fit. A worsen of model fit between Aims 3 and 4 (as indicated by an increase in AIC, BIC, or ssa-BIC of values >10) would provide support for superiority of the 'psychological-coping' mechanism as opposed to the 'availability-affordability' mechanism.

All data and code for this analysis is available on the Open Science Framework.

## Results

### Aim 1

Two-thirds of adults (67.1%; n = 944) at Wave 2 identified as current drinkers. Table 2 presents the estimated proportions of current drinkers meeting the threshold for at-risk drinking, and the mean AUDIT-C score, at Wave 2 (reference period pre-pandemic level) compared to Wave 3 (i.e., during lockdown, March-July 2020). The proportion of drinkers who satisfied the criteria for at-risk drinking decreased slightly from 35.9% (95%CI 32.8–39.0) before the pandemic to 32.0% (95%CI 28.6–35.5) during the lockdown. The AUDIT-C mean scores were similar during these two time periods, and the equality test indicated that there were no significant difference, indicating that the mean level of alcohol use remained stable from pre-pandemic levels into the first four months of the lockdown.

### Aim 2

The results for CFA are presented in Table 3. The *a priori* three-factor CFA model was supported: the CFI and TLI values were >0.98, the RMSEA was <0.06. Each factor was measured by positive, strong, and statistically significant factor loading, and moderate-high correlations were observed.

The standardized regression coefficients for the three factor scores and selected socio-economic, alcohol use, mental health, and psychological characteristics, are presented in Table 4.

Important differences emerged between the respondent characteristics and the drinking motive factors. Adult drinkers who had higher levels of drinking pre-pandemic ($\beta$ = 0.117, p<0.001), those who reported over-purchasing alcohol during the pandemic ($\beta$ = 0.217, p<0.001), those meeting caseness for depression or anxiety at the start of the pandemic ($\beta$ = 0.182, p<0.001), and those with higher levels of paranoia ($\beta$ = 0.016, p<0.05), death anxiety ($\beta$ = 0.005, p<0.05), and extraversion ($\beta$ = 0.031, p<0.01), but lower levels of resilience ($\beta$ = -0.015, p<0.05) had higher estimates of *Coping* motives.

Younger drinkers ($\beta$ = -0.006, p<0.001), those with higher pre-pandemic levels of drinking ($\beta$ = 0.066, p<0.001), those reporting increased purchasing of alcohol during the pandemic ($\beta$ = 0.169, p<0.001), those meeting PTSD caseness criteria ($\beta$ = 0.312, p<0.01), and those with higher levels of extraversion ($\beta$ = 0.053, p<0.001) and death anxiety ($\beta$ = 0.005, p<0.05), all had higher estimates of drinking for *Social Enhancement* motives.

**Table 2. Tests of the proportion of people indicating risky drinking pattern (as measured by AUDIT-C threshold of 5+) and mean drinking score from AUDIT-C reflecting change from pre-pandemic levels (measured at Wave 2) to during lockdown (March-July 2020; Wave 3) (N = 944).**

| | Wave 2 | Wave 3 | Null model | | | Alternative | | | Null vs. alternative model |
|---|---|---|---|---|---|---|---|---|---|
| | | | AIC | BIC | Ssa-BIC | AIC | BIC | Ssa-BIC | LRT |
| | % (95% CI) | % (95% CI) | | | | | | | |
| AUDIT-C (Risky drinking 5 + threshold) | 35.9 (32.8, 39.0) | 32.0 (28.6, 35.5) | 1852.742 | **1872.143** | 1859.439 | **1849.132** | 1873.383 | **1857.503** | $\chi^2$ = 5.44, df = 1, p = .02 |
| | Mean (se) | Mean (se) | | | | | | | |
| AUDIT-C (Mean score) | 4.246 (.07) | 4.155 (.08) | 6507.567 | **6526.968** | **6514.264** | **6507.396** | 6531.646 | 6515.767 | $\chi^2$ = 2.12, df = 1, p = .14 |

**Table 3. Factor loadings from the measurement model of 15 drinking motive indicators of past week drinking–COVID-19 Psychological Research Consortium (C19PRC) Study UK Strand, Wave 2 (April-May, 2020).**

| *Over the past week, how often have you consumed alcohol for the following reasons?* | Social Enhancement Factor | Coping Factor | Conformity Factor | Weighted frequency Yes (%) |
|---|---|---|---|---|
| To have fun (DM1) | 0.853 (0.033) | | | 55.6% |
| To enjoy a special occasion (DM7) | 0.728 (0.033) | | | 47.1% |
| To celebrate (DM13) | 0.834 (0.035) | | | 36.4% |
| Out of habit (DM2) | | 0.677 (0.031) | | 49.2% |
| To relax (DM3) | | 0.633 (0.045) | | 77.6% |
| To help you sleep (DM4) | | 0.857 (0.023) | | 25.8% |
| To feel less stressed (DM5) | | 0.877 (0.019) | | 42.0% |
| To cheer yourself up (DM9) | | 0.855 (0.018) | | 50.4% |
| To reward yourself (DM10) | | 0.699 (0.031) | | 57.5% |
| To feel like the old you (DM11) | | 0.848 (0.021) | | 31.6% |
| To forget problems (DM12) | | 0.901 (0.019) | | 29.0% |
| Because you were bored (DM15) | | 0.895 (0.017) | | 60.3% |
| Because you fear of missing out (DM14) | | | 0.984 (0.013) | 14.7% |
| Because you felt pressured by family/friends/peers (DM6) | | | 0.919 (0.018) | 14.7% |
| Because you felt left out (DM8) | | | 0.976 (0.013) | 14.0% |
| Factor correlations | | | | |
| Coping Factor | 0.642 | | | |
| Conformity Factor | 0.845 | 0.922 | | |
| Chi-square (*df*) *p* | 334.796 (*87*) *p* < .001 | | | |
| CFI | 0.984 | | | |
| TLI | 0.981 | | | |
| RMSEA (90% CI) | 0.055 (0.49–0.61) | | | |

Adult drinkers with higher estimates of drinking for *Conformity* motives were more likely to be a key-worker (β = 0.130, p<0.05), low income earners (compared to those in the second highest income bracket) (β = -0.166, p<0.05), have higher levels of pre-pandemic drinking (β = 0.092, p<0.001), meeting PTSD caseness criteria (β = 0.304, p<0.001), higher levels of paranoia (β = 0.015, p<0.05), death anxiety (β = 0.005, p<0.05), higher levels of extraversion (β = 0.040, p<0.001), have increased their purchasing of alcohol during the pandemic (β = 0.217, p<0.001), and have lower levels of neuroticism (β = -0.031, p<0.05).

### Aim 3

Pre-pandemic drinking level (AUDIT-C scores, Wave 2) predicted drinking level during lockdown (AUDIT-C scores; Wave 3) (β = 0.294, p < .001). The regression pathway of the Wave 3 AUDIT-C score on *Coping* (β = 0.713, p<0.05) was statistically significant, but not for *Social Enhancement* (β = -0.018, p = 0.946) or *Conformity* (β = -0.190, p = 0.745). The regression pathways of *Coping* on solitary drinking (β = 0.193, p<0.001) and *Social Enhancement* to solitary drinking (β = -0.148, p<0.001) were statistically significant (but not for *Conformity*, β = -0.118, p = 0.132). The indirect effect of the AUDIT-C score (Wave 3) on *Coping* via solitary drinking was not statistically significant (β = 0.034, p = .167). [overall model fit: AIC = 8908.269; BIC = 9344.590; ssa-BIC = 9058.756].

### Aim 4

Neither path between income loss, or reduced working hours, and the AUDIT-C score (Wave 3) was statistically significant (β = -0.018, *p* = 0.448 and β = 0.023, *p* = 0.415, respectively)

**Table 4. Estimated effects of respondent characteristics on drinking motive factor scores (social enhancement, coping, and conformity) as assessed at Wave 2, COVID-19 Psychological Research Consortium (C19PRC) Study UK Strand, Wave 2 (April-May 2020).**

| Socio-demographic, alcohol use, mental health, and psychological characteristics of respondents | | Social Enhancement | Coping | Conformity |
|---|---|---|---|---|
| Gender (Wave 1) | Females | -0.031 (0.047) | 0.027 (0.046) | -0.007 (0.045) |
| | Males (R) | | | |
| Age (years; Wave 1) | | -0.006 (0.002)*** | 0.002 (0.002) | -0.002 (0.002) |
| 2019 Household income (Wave 1) | ≤£15.490 (R) | | | |
| | £15,491-£25,340 | -0.078 (0.083) | -0.058 (0.088) | -0.048 (0.082) |
| | £25,341-£38,740 | 0.013 (0.086) | -0.107 (0.090) | -0.044 (0.084) |
| | £38,741-£57,903 | -0.117 (0.083) | -0.178 (0.087)* | -0.166 (0.081)* |
| | ≥£57,931 | -0.072 (0.084) | -0.157 (0.088) | -0.120 (0.081) |
| Lost income due to pandemic (March-May 2020; Wave 2) | Yes | -0.060 (0.049) | 0.009 (0.049) | -0.034 (0.047) |
| | No (R) | | | |
| Reduced work hours due to pandemic (March-May 2020; Wave 2) | Yes | 0.032 (0.051) | 0.007 (0.050) | 0.042 (0.049) |
| | No (R) | | | |
| Key-worker (Wave 2) | Yes | 0.068 (0.052) | 0.096 (0.049) | 0.103 (0.049)* |
| | No (R) | | | |
| Ethnicity (Wave 1) | White | -0.166 (0.103) | 0.008 (0.110) | -0.061 (0.101) |
| | Other (R) | | | |
| Living alone (Wave 1) | Yes | -0.125 (0.066) | 0.030 (0.065) | -0.028 (0.063) |
| | No (R) | | | |
| Children (<18 years) in household (Wave 1) | Yes | 0.047 (0.056) | 0.098 (0.054) | 0.109 (0.054)* |
| | No (R) | | | |
| Retrospective recall pre-pandemic AUDIT-Score (Wave 2) | | 0.066 (0.012)*** | 0.117 (0.012)*** | 0.092 (0.011)*** |
| Increased purchasing of alcohol (past four weeks; Wave 2) | | 0.169 (0.027)*** | 0.217 (0.026)*** | 0.217 (0.025)*** |
| Depression or generalised anxiety caseness (Wave 1) | Yes | -0.013 (0.061) | 0.182 (0.070)* | 0.100 (0.064) |
| | No | | | |
| PTSD caseness (Wave 1) | Yes | 0.312(0.085)*** | 0.133 (0.088) | 0.304 (0.088)*** |
| | No | | | |
| Loneliness caseness (Wave 1) | Yes | 0.024 (0.054) | 0.059 (0.055) | 0.056 (0.053) |
| | No | | | |
| Paranoia (Wave 1) | | 0.011 (0.007) | 0.016 (0.007)* | 0.015 (0.006)* |
| COVID-19 anxiety (Wave 1) | | 0.000 (0.001) | 0.002 (0.001) | 0.001 (0.001) |
| Death anxiety (Wave 1) | | 0.005 (0.002)* | 0.005 (0.002)* | 0.005 (0.002)* |
| Intolerance of uncertainty (Wave 1) | | 0.005 (0.004) | 0.006 (0.004) | 0.005 (0.004) |
| Extraversion (Wave 1) | | 0.053 (0.012)*** | 0.031 (0.012)** | 0.040 (0.011)*** |
| Neuroticism (Wave 1) | | -0.030 (0.016) | -0.020 (0.016) | -0.031 (0.015)* |
| Resilience (Wave 1) | | 0.000 (0.007) | -0.015 (0.006)* | -0.010 (0.006) |

Note.

*** $p < .001$

** $p < .01$

* $p < .05$.

[overall model fit: AIC = 8911.400; BIC = 9357.416; ssa-BIC = 9065.231]. Testing these two additional pathways had a marginal impact on the model fit (i.e., only the BIC value obtained from the model estimated at Aim 3 compared to Aim 4 differed by >10, which suggests a worsening of model fit).

## Discussion

The unique nature of the COVID-19 pandemic, with its abrupt and widespread impact on all facets of life, ensured that it would be difficult to extrapolate from studies conducted during recent periods of socio-economic downturn or other traumatic events/natural disasters, to predict whether, and how, adults' alcohol use patterns might change during this tremendous period of upheaval and uncertainty. Here, we analysed rich, longitudinal survey data to test alternative drivers–namely individual-level psychological and socio-economic factors–of alcohol consumption patterns among adults living in the UK during the first national lockdown in spring 2020.

The main findings can be summarised succinctly. First, the proportion of adults meeting the threshold for 'at-risk' drinking (as assessed by an AUDIT-C score of 5+) decreased significantly during the lockdown compared to retrospectively recalled pre-pandemic levels (i.e., from 35.9% to 32.0%), but overall mean levels of drinking remained stable. Second, neither loss of income nor reduced working hours during the first two months of the spring 2020 UK lockdown predicted drinking motives during this period. Third, increased purchasing of alcohol in the first month of lockdown was positively associated with all three drinking motives during the lockdown. Fourth, drinking to cope motives were uniquely associated with experiences of depression or anxiety and lower levels of resilience; other experiences of mental health distress (e.g., PTSD caseness, increased levels of paranoia and death anxiety) were associated with social enhancement and/or conformity drinking motives. Fifth, only drinking to cope motives predicted both solitary drinking and drinking levels during lockdown, but the indirect effect of drinking to cope motives predicting increased drinking via engaging in solitary drinking was not statistically significant. Finally, drinking levels during lockdown were not predicted by social enhancement or conformity drinking motives, loss of income or reduced working hours earlier in the pandemic.

Overall, evidence in support of the 'affordability-availability' mechanism was weak. The proportion of adults engaging in 'at-risk' drinking declined significantly, but only by 3.9%, during lockdown. It is likely that, as posited by Rehm, Kilian [8], this decrease may have been driven by a reduction in heavy episodic drinking events. However, we failed to provide support for the hypothesis that a decline in drinking levels would be predicted by loss of income or reduced working hours during the earliest phases of lockdown. On one level, this finding is not entirely surprising. Widespread comparative analyses of the current crisis and that of the recent 2007–09 Great Recession has highlighted major differences between the national and international fiscal policy responses implemented to manage these economic crises [56]. Whereas the Great Recession was characterised by a major reduction in government expenditure, the COVID-19 crisis has seen the swift implementation of multiple stimulus packages to provide support for employers and employees who were hit financially by the sudden and widespread closures of business and workplaces [57].

Overall, it seems more plausible that reduced working hours or loss of income might influence drinking levels indirectly by individuals struggling to adapt to boredom or disruption to normal daily routines, due to not working [6]. However, we found no evidence that reduced working hours or loss of income influenced drinking motives, be it for coping, social enhancement, or conformity.

Stronger support for the 'psychological' mechanism emerged in this study. Consistent with some of our *a priori* predictions, and previous research [24, 27], experiences of depression or anxiety and low levels of resilience were uniquely associated with drinking to cope motives during lockdown. A well-established finding in the literature is the cyclic nature of the relationship between negative affect and alcohol consumption: higher levels of alcohol use are

typically precipitated by feelings of low mood/experiences of anxiety, but then increased alcohol use further exacerbates negative affect leading to increased drinking [58]. It will be important to continue to assess how adults who engaged in drinking to cope during the earliest stages of lockdown manage their drinking as the pandemic unfolds, as on-going self-isolation for these individuals may be associated with a deterioration in mental health and subsequent increases in alcohol use and related harmful consequences [8].

The finding that higher levels of death anxiety, but not anxiety about COVID-19 more generally, was associated with all three drinking motives is noteworthy. Whilst the latter finding largely concurs with Wardell, Kempe [24]'s work on coping motives, the former is novel, but not an unexpected association during the earliest weeks of the pandemic when daily death tolls were increasing rapidly. Together, these associations indicate that adult drinkers who were susceptible to drinking to cope were likely struggling with additional emotional burdens unique to the pandemic, in additional to experiences of anxiety or depression typically characteristic of a 'drinking to cope' relationship with alcohol.

That drinking to cope motives only (not social enhancement or conformity motives) were associated with increased drinking levels during the lockdown provides further evidence in support of the psychological-coping mechanism. Whilst these findings runs counter to recent evidence which suggests that both enhancement and social motives predicted lower alcohol consumption during lockdown among heavy college drinkers [17], they are consistent with other studies which demonstrated that social drinking motives were not associated with greater alcohol consumption during the two months of lockdown in the UK [25]. Considered alongside the findings relating to over-purchasing of alcohol, it is plausible that drinkers who were motived to drink to cope ended up consuming more of the alcohol they had 'stock-piled', whereas individuals who drank for social enhancement or conformity motives, who also engaged in over-purchasing, did not. It appears likely that individuals in this study who drank for social enhancement or conformity motives during the lockdown continued to use alcohol in a normal pattern, albeit in a different social context.

Our findings in relation to associations between PTSD and drinking for social enhancement or conformity motives were not predicted and are difficult to explain. Whilst there is a sizeable literature demonstrating a link between PTSD, drinking to cope motives, and increased alcohol use, the evidence for PTSD influencing drinking for other motives is sparse and mixed [59]. Moreover, there has been some debate as to what PTSD means in the context of COVID-19 [60–62], and hence it may be less feasible to extrapolate from the existing evidence base on the link between PTSD and drinking motives. Forced to speculate, there might be something unique about the experience of processing feelings of distress relating to the threat of the coronavirus (compared to other traumatic events) that motived adults to use alcohol in ways that brought them together with social support networks to process the distress in an external way, as opposed to drinking to numb or cope with internalised distress. As yet, however, this hypothesis requires testing should this somewhat novel finding be replicated in other studies.

Several study strengths and limitations are noted. The C19PRC Study sample was obtained via non-probability quota sampling methods and was established in March 2020, and therefore a true 'pre-pandemic' baseline of normal alcohol use was not available. Adults aged 18 years and older were asked about their usual 'pre-pandemic' drinking at Wave 2, at which stage ~30% of the baseline sample had been lost to follow-up, and 'during-pandemic' drinking at Wave 3, at which point a further 10% of the sample had been lost. We have demonstrated previously that respondents retained at follow-up were more likely to be male, older, higher income earners, of White ethnicity, living outside cities, living in adult-only households, and to have been born/raised in the UK [32]. However, sampling weights and/or robust maximum

likelihood estimation were employed to maximise the utility of available data. Approximately one-third of the Wave 2 sample (32.9%) were classified as non-current drinkers, which is lower, but in general comparable, to estimates obtained from other UK surveys reporting trends for non-current drinking among adults aged 16 years and older (e.g., ~43% in the 2017 Opinions and Lifestyle Survey) [63]. All mental health assessments (e.g., anxiety, depression, PTSD) were based on self-report rather than clinician administered interviews, and this may have resulted in over- or under-estimation of disorder caseness. A set of 15 statements was administered to measure drinking motives during the pandemic, but there was a particular focus on measuring drinking to cope motives given the unprecedented change to daily life activities during lockdown. Due to survey constraints (i.e., financial resources and in an attempt to minimise respondent burden), other traditionally assessed drinking motives— social, enhancement, and conformity motives—were measured using a small number of items (i.e., six items to measures these three constructs). Items assessing social and enhancement motives were combined to facilitate the specification and estimation of the confirmatory factor model, and this model provided a good fit to the data.

The analytic framework for this study was guided by theory and emerging research evidence in this area; however, we did not collect data on other variables relating to alcohol use that may have explained changes in consumption. For example, some adults in the UK appear to have used the lockdown as an opportunity to 'audit' or 're-assess' their relationship with alcohol and to pilot periods of abstinence away from external pressures to drink heavily [29]. Alternative models of the associations between the main study variables are also possible. It will be important to replicate the findings reported here in similarly large and diverse samples (e.g., including those more vulnerable groups of society prone to harmful alcohol-related consequences).

## Conclusion

Our findings, which both contribute to, and extend, the emerging body of evidence in this area suggest that, in the earliest stage of the pandemic, evidence in support of psychological mechanisms driving changes in adults' alcohol use is stronger than that offered by an 'affordability-availability' mechanism perspective.

## Author Contributions

**Conceptualization:** Orla McBride, Oisín Harkin, Mark Shevlin.

**Data curation:** Sarah Butter.

**Formal analysis:** Orla McBride.

**Funding acquisition:** Orla McBride, Mark Shevlin, Jamie Murphy, Liam Mason, Todd K. Hartman, Liat Levita, Kate M. Bennett, Jilly Gibson-Miller, Richard P. Bentall.

**Investigation:** Orla McBride, Mark Shevlin, Richard P. Bentall.

**Methodology:** Orla McBride, Sarah Butter, Mark Shevlin, Jamie Murphy, Liam Mason, Todd K. Hartman, Ryan McKay, Philip Hyland, Liat Levita, Kate M. Bennett, Thomas V. A. Stocks, Jilly Gibson-Miller, Anton P. Martinez, Frédérique Vallières, Richard P. Bentall.

**Project administration:** Sarah Butter.

**Supervision:** Mark Shevlin, Richard P. Bentall.

**Validation:** Orla McBride.

**Visualization:** Orla McBride.

**Writing – original draft:** Orla McBride.

**Writing – review & editing:** Orla McBride, Eimhear Bunting, Oisín Harkin, Sarah Butter, Mark Shevlin, Jamie Murphy, Liam Mason, Todd K. Hartman, Ryan McKay, Philip Hyland, Liat Levita, Kate M. Bennett, Thomas V. A. Stocks, Jilly Gibson-Miller, Frédérique Vallières, Richard P. Bentall.

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
