## [Decision Letter · Decision Letter 0]

15 Dec 2021

PONE-D-21-30816Testing both affordability-availability and psychological-coping mechanisms underlying changes in alcohol use during the COVID-19 pandemicPLOS ONE

Dear Dr. McBride,

Thank you for submitting your manuscript to PLOS ONE. After careful consideration, we feel that it has merit but does not fully meet PLOS ONE’s publication criteria as it currently stands. Therefore, we invite you to submit a revised version of the manuscript that addresses the points raised during the review process.

We look forward to receiving your revised manuscript.

Kind regards,

Gabriel A. Picone

Academic Editor

PLOS ONE

Journal Requirements:

-https://osf.io/w3anh/

The text that needs to be addressed involves the abstract.

In your revision ensure you cite all your sources (including your own works), and quote or rephrase any duplicated text outside the methods section. Further consideration is dependent on these concerns being addressed.

Reviewers' comments:

Reviewer's Responses to Questions

**Comments to the Author**

1. Is the manuscript technically sound, and do the data support the conclusions?

Reviewer #1: Yes

2. Has the statistical analysis been performed appropriately and rigorously? 

Reviewer #1: I Don't Know

3. Have the authors made all data underlying the findings in their manuscript fully available?

Reviewer #1: Yes

4. Is the manuscript presented in an intelligible fashion and written in standard English?

Reviewer #1: Yes

5. Review Comments to the Author

Reviewer #1: This is my second time reviewing this article. As I stated the first time around, there is a lot to like about this paper, including a large sample size, interesting and important questions, a great summary of prior work on changes in alcohol consumption as a result of the pandemic, and public access to the data. I offered the following suggestions to strengthen the paper when I reviewed this the first time, and the authors were highly responsive to all of my concerns in this updated version. I have one more question, though. Typically, there are four drinking motives, and I wondered why the authors combined social and enhancement? Overall, I think this paper makes a nice contribution to the literature.

--Prior concerns that were adequately addressed--

1. The paper would benefit from a figure or figures. There are a lot of variables and paths being tested and a figure would help in clarifying the models and results.

2. I had a hard time seeing how the four aims mapped onto testing the “psychological” and “affordability-availability” mechanisms. For instance, wouldn’t it be illuminating to test whether drinking to cope motives explain the link between increases in variables like depression, loneliness etc and increases in drinking behavior as a result of the pandemic? Also, did the authors control for other motives in their analyses to see unique effects? Finally, it was unclear why some of the variables were included (e.g., paranoia, personality traits). It would be helpful to better motivate these in the introduction.

3. I understand that there are 3 waves of data, but it was unclear when some of the measures were assessed (this info is omitted for many of the measures in the methods section). It would be helpful to clarify which measures were assessed when and, in the case where measures were collected at multiple time points, an explanation is needed for choosing one particular wave over another.

4. Relatedly to my point above, the authors don’t seem to take advantage of the longitudinal nature of the data in their analyses. The title of the paper and hypotheses focus on changes in alcohol use, but the analyses don’t account for changes in drinking behavior pre- to post-pandemic, or this wasn’t clear.

5. The sample is described as being nationally-representative on page 9, but on page 20 a limitation is that the results can’t be generalized to all adults. Are the authors able to clarify?

6. The small decrease in at risk drinking is described as being “slight” in the results section and the confidence intervals overlap, but the difference is described as being significant in some places of the text. Are the authors able to clarify this, too?

7. It’s unclear if you were able to account for individuals who stopped drinking as a result of the pandemic in analyses.

8. I thought scores on the AUDIT-C indicative of hazardous drinking varied for men (score of 4) and women (score of 3). The authors used a score of 5, but perhaps I’m not aware of this convention.

6. PLOS authors have the option to publish the peer review history of their article (what does this mean?). If published, this will include your full peer review and any attached files.

Reviewer #1: No

---

## [Author Response · Author response to Decision Letter 0]

5 Feb 2022

All issues raised have been responded to in the 'Response to Reviewers' Document

---

## [Decision Letter · Decision Letter 1]

24 Feb 2022

Testing both affordability-availability and psychological-coping mechanisms underlying changes in alcohol use during the COVID-19 pandemic

PONE-D-21-30816R1

Dear Dr. McBride,

We’re pleased to inform you that your manuscript has been judged scientifically suitable for publication and will be formally accepted for publication once it meets all outstanding technical requirements.

Kind regards,

Gabriel A. Picone

Academic Editor

PLOS ONE

Additional Editor Comments (optional):

Reviewers' comments:

Reviewer's Responses to Questions

**Comments to the Author**

1. If the authors have adequately addressed your comments raised in a previous round of review and you feel that this manuscript is now acceptable for publication, you may indicate that here to bypass the “Comments to the Author” section, enter your conflict of interest statement in the “Confidential to Editor” section, and submit your "Accept" recommendation.

Reviewer #1: All comments have been addressed

2. Is the manuscript technically sound, and do the data support the conclusions?

Reviewer #1: (No Response)

3. Has the statistical analysis been performed appropriately and rigorously? 

Reviewer #1: (No Response)

4. Have the authors made all data underlying the findings in their manuscript fully available?

Reviewer #1: (No Response)

5. Is the manuscript presented in an intelligible fashion and written in standard English?

Reviewer #1: (No Response)

6. Review Comments to the Author

Reviewer #1: (No Response)

7. PLOS authors have the option to publish the peer review history of their article (what does this mean?). If published, this will include your full peer review and any attached files.

Reviewer #1: No

---

## [Editor Report · Acceptance letter]

3 Mar 2022

PONE-D-21-30816R1 

Testing both affordability-availability and psychological-coping mechanisms underlying changes in alcohol use during the COVID-19 pandemic 

Dear Dr. McBride:

I'm pleased to inform you that your manuscript has been deemed suitable for publication in PLOS ONE. Congratulations! Your manuscript is now with our production department. 

Kind regards, 

on behalf of

Dr. Gabriel A. Picone 

Academic Editor

PLOS ONE